# Investigation of the Change in Roughness and Microhardness during Laser Surface Texturing of Copper Samples by Changing the Process Parameters

**Risham Singh Ghalot** [1], **Lyubomir Lazov** [1,*], **Emil Yankov** [2,*] **and Nikolay Angelov** [3]

1   Faculty of Engineering, Rezekne Academy of Technology, LV-4601 Rēzekne, Latvia; singhrisham93@gmail.com
2   Materials Science and Technology, University of Ruse "Angel Kanchev", 7017 Ruse, Bulgaria
3   Department of Physics, Chemical and Ecology, Technical University of Gabrovo, 5300 Gabrovo, Bulgaria; angelov_np@abv.bg
*   Correspondence: lyubomir.lazov@rta.lv (L.L.); eyankov@uni-ruse.bg (E.Y.); Tel.: +359-888869092 (L.L.); Tel.: +359-895614247 (E.Y.)

**Abstract:** The aim of this research is to achieve a high-quality and long-lasting laser marking of ammunition, which is of interest to the defense industry. The study is about the effects of speed, raster pitch and power on the roughness and microhardness of the marked areas of copper samples. The experiments were carried out with a fiber laser and a copper bromide laser—modern lasers widely used in industrial production. Laser power, scan speed and raster step were varied to determine their effects on the resulting microhardness and surface roughness. The lasers operate in different wavelength ranges, with the optical laser operating at 1064 nm in the near-infrared region and the copper bromide laser at 511 nm and 578 nm in the visible region, allowing the influence of wavelengths on the process to be investigated. The roughness and microhardness velocity dependence for three powers and two pulse durations for the fiber laser were obtained from the experimental data. The dependence of roughness and microhardness on the raster step for both types of lasers was also demonstrated.

**Keywords:** laser marking; roughness; copper; laser surface texturing; raster step; microhardness measurement

## 1. Introduction

Copper and its alloys are some of the most widely used materials in industry, with applications in the production of ammunition, non-magnetic tools and equipment, architectural elements, sculptures, products requiring high thermal conductivity, electrical and electronic products, etc. [1–3].

Laser surface texturing is one of the most innovative material processing technologies nowadays, where new surface properties are obtained, and with significant applications in tribology, medicine, aeronautics, etc. [4–8]. Laser texturing technology has a number of advantages, such as non-contact processing, precision, reproducibility, high productivity and environmental friendliness [9–11].

Among several techniques for laser-induced texturing of metal surfaces, the attention of researchers has recently focused on the laser surface melting (LSM) technique, inducing significant changes in the microstructure [12–17].

In a number of articles, the authors research the influence of a number of technological parameters (average power, pulse sequence frequency, scan speed, defocus, step between lines, etc.) on the laser texturing process [18–22].

In their publication, M. Naumova and I. Morozova [18] studied the influence of technological parameters during laser texturing on the surface of brass samples for the purpose of forming an oxide layer and color marking. The influence of technological



parameters, such as average power, pulse frequency and scan speed, on the depth of roughness during the technological process of titanium samples was studied by C. Velotti et al. [19]. They used a Yb: YAG fiber laser to conduct their experiments. The depth of roughness was also investigated by D. Gogoi in his master's thesis, where he uses a $CO_2$ laser, varying the power and scan speed. In publication [20], the author investigated the influence of speed, the number of repetitions and the volumetric density of absorbed energy on the process of laser marking of stainless-steel samples. In article [21], the researcher analyzed laser texturing of the surface of copper samples by achieving super hydrophobicity based on certain surface modifications and roughness. The author of a master's thesis, [22], performed laser engraving with a $CO_2$ laser and concluded that the laser power, scanning speed and number of dots per inch have convincing effects on the samples. In the experiments, it was observed that, as the laser power increased, the surface roughness increased, while lower and mid-range laser power resulted in better surface quality with lower roughness.

In [23], the authors used a dual-frequency Nd: YAG green laser for laser texturing while considering the crucial influence of the scanning speed on the surface roughness. The authors of [24] performed laser texturing of several materials to study the dependence of the process on the absorption of laser radiation and the thermal properties of the material. It was unequivocally shown that the processing of materials strongly depends on the absorption of laser radiation and the thermal properties of the materials. Of all the laser machined materials (copper, aluminum, steel and titanium), copper is the most difficult to machine due to its high reflectivity. This imposes and requires the highest laser power in the processing area (from 1 kW at a processing speed of 0.1 m/min).

A number of studies also consider issues related to the change in microhardness in the processing area after laser impact. The authors of [25] investigated the increase in hardness of a 6 mm thick mild steel substrate after laser processing with 30 W and a fiber laser (1060 nm) with a beam diameter of 0.4 mm. By laser processing the substrate with a laser power of 21 W, a 100 μm raster step, 200 kHz frequency and a scan speed of 40 mm/s and analyzing it with a load of 0.5 kg and hold of 10 s, the highest microhardness achieved for mild steel was 281.72 HV. The marked zone of greater microhardness has a martensitic grain structure, which gives an increase of 86% in the surface microhardness of the substrate. In the analysis in [26], the researchers found that the laser beam treatment introduces a self-hardening effect on the copper surface, due to which the microstructure of the copper surface changes and an increase in hardness occurs. In addition, laser processing caused rapid surface melting and recrystallization of the surface of the copper specimen, resulting in lower roughness and higher microhardness and corrosion resistance effects in the nature of the material. The results obtained in [27] show that the change in the overlap of the laser beam leads to an increased hardness of the copper surface.

As a result of this analysis, we can state that the publications on laser texturing and marking of copper products are extremely few in number.

For this reason, the aim of our research was to contribute to a more in-depth study of the process of laser texturing of copper samples by analyzing the influence of some basic parameters, such as laser radiation power, processing speed, raster step and the influence of wavelengths for two types of lasers (a fiber laser and a copper bromide laser) on roughness and microhardness.

## 2. Materials and Methods

The methodology used here makes it possible to analyze the relationship between the contrast, roughness and microhardness of some parameters, such as scanning speed ($v$), raster step ($\Delta x$), power ($P$) and pulse duration ($\tau$). The sequence for planning and performing the experiments is as follows:

(a)  Setting tasks to be performed in the present experiments:

- Determination of roughness dependence on velocity—$R_a = R_a\ (v)$—for a fiber laser for two pulse durations.

- Determining the dependence of microhardness on speed—$HV = HV(v)$—for a fiber laser for two pulse durations.
- Determining the dependence of roughness on the raster step—$R_a = R_a(\Delta x)$—for a fiber laser and a copper bromide laser.
- Determining the dependence of microhardness on the raster step—$HV = HV(\Delta x)$—for a fiber laser and a copper bromide laser.

(b)   Designing matrices for performing the experiments:

For each task, a separate matrix is made with changes to the various technological parameters. An example matrix is shown in Figure 1. It is suitable for the implementation of tasks 1 and 2. The rows provide different velocity values for each square. Different power values are provided along the columns. Matrixes of similar structure are made for tasks 1–4 (see Figures 1–3).

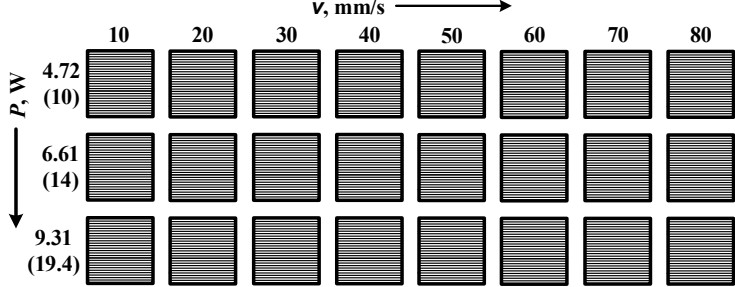

**Figure 1.** Matrix for performing experiments with a fiber laser for $\tau = 100$ ns and $\tau = 200$ ns at varying marking speed and average power.

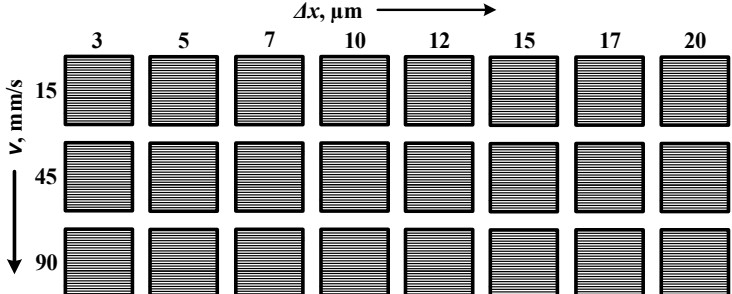

**Figure 2.** Matrix for performing experiments with a fiber laser for $\tau = 100$ ns at varying marking speed and raster step.

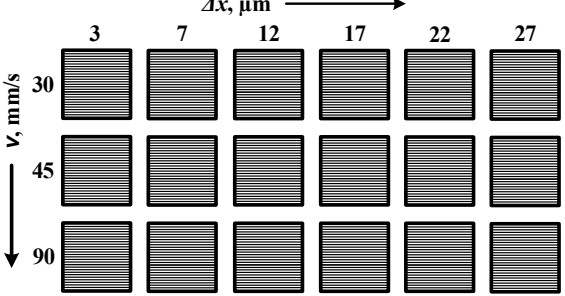

**Figure 3.** Matrix for performing experiments with a CuBr laser for $\tau = 30$ ns at varying marking speed and raster step.

(c)   Performing experiments and marking of samples:

The marked areas are square shaped with dimensions of 10 mm × 10 mm with 2 mm distance between them. The copper plates are marked with a Rofin F20 fiber laser (1064 nm) and a CuBr laser (511 and 578 nm). During maceration with a Rofin

F20 fiber laser on 3 copper plates (Figure 1), 24 planned experiments were carried out with a changing speed from 10 to 80 mm/s in a step of 10 mm/s and an average power of 4.72 W, 6.61 W and 9.31 W, with constant parameters being the raster pitch, focal length 184 mm, frequency 20 kHz and pulse duration 100 ns, with a pulse duration of 200 ns for the repeated experiment. When marking the experimental matrix (Figure 2) with a Rofin F20 fiber laser on the copper plate, 24 planned experiments were carried out with a changing marking speed of 15 mm/s, 45 mm/s and 90 mm/s and a raster step of 3 μm, 5 μm, 7 μm, 10 μm, 12 μm, 15 μm, 17 μm and 20 μm, with the constant parameters being average power, focal length 183 mm, frequency 20 kHz and pulse duration 100 ns. For the laser marking of 3 copper sheets with a CuBr laser, the experimental matrix consisted of 18 experiments with memorized parameters of marking speeds of 30 mm/s, 45 mm/s and 90 mm/s and a raster step of 3 μm, 7 μm, 12 μm, 17 μm, 22 μm and 27 μm, with constant parameters being an average power of 10 W, a focal length of 300 mm, frequency of 20 kHz and pulse duration of 30 ns.

(d)    Roughness measurement (for each square):

Roughness measurements and the resulting microstructure were examined with an Olympus model "OLS5100-EAF" laser microscope. The obtained microstructural images were carried out using a 20× objective, with a magnification of 451×, as the examined area for each measurement was 644 × 644 μm, with a measurement accuracy of ±0.03 μm.

From the obtained 3D images with the laser system of the microscope, the roughness $R_a$ and $R_z$ perpendicular to the marking lines with a length of 644 μm and the roughness $R_q$ for the entire examined area, 644 × 644 μm, were measured. The obtained values are plotted in tables and graphical dependencies of changes in roughness depending on speed and raster step during surface laser processing are shown. The built dependencies are presented in the results.

(e)    Microhardness measurement (for each square):

The hardness measurement was carried out according to the Vikes method with Inovatest "Nexus 4000-4302"; its capabilities are presented. The choice of load was established according to microscopic analysis and surface roughness. The maximum load is 1.961 N, with 5 consecutive measurements performed in a 200 μm step in the marked area. Measurements took place at a microscopic zoom of 40×. The measured values are tabulated and averaged over the five measurements of each mark. With the obtained data, graphical dependencies were built for the influence of speed, pitch and mass on the roughness and hardness. Comparison charts were also constructed and are presented in Section 3 results.

## 3. Results

The marked substrates of Cu by fiber and copper-bromide laser were analyzed under a laser microscope and hardness tester for average roughness and micro-hardness, respectively. To determine the change in roughness ($R_a$) and hardness ($HV$) after laser marking, they were measured in the delivery condition (Figure 4). The initial roughness in the condition of delivery of the copper plates is in the range of 0.22 μm to 0.64 μm. The measured hardness as delivered (before laser treatment) for the copper plates is in the range of 61.2 HV to 67.1 HV.

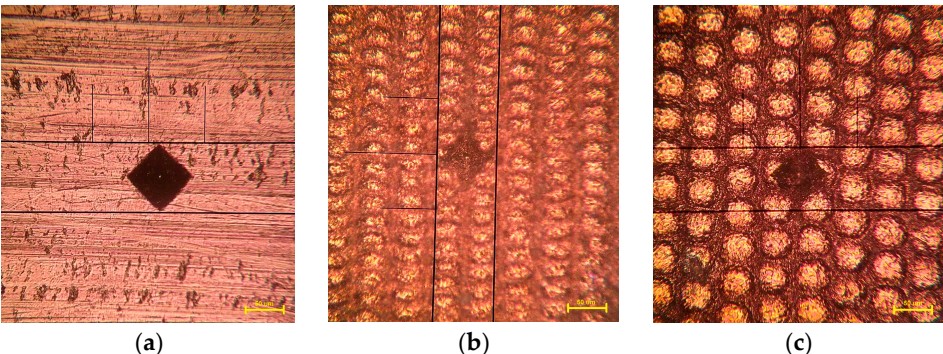

|  |  |  |
|:-:|:-:|:-:|
| (**a**) | (**b**) | (**c**) |

**Figure 4.** Images at $451\times$ magnification of (**a**) microstructure of the measured hardness of an untreated surface, (**b**) measured hardness on a laser-marked surface with an overlap factor, and (**c**) measured hardness on a laser-marked surface without overlap.

### 3.1. Determination of Roughness Dependence on Velocity—$R_a = R_a(v)$—For a Fiber Laser for Two Pulse Durations

During the experiments, the speed $v$ varied from 10 mm/s to 80 mm/s. The studies were at two pulse durations, 100 ns and 200 ns. The following parameters were kept constant: frequency 20 kHz and raster step 10 μm. A comparison of the marked surfaces (highest and lowest roughness) can be observed in Figures 5 and 6. The graphical images in Figure 7 depict the obtained analytical values of the average roughness of the Cu after the laser treatment for a pulse duration $\tau$ of 100 ns (Figure 7a) and 200 ns (Figure 7b). The graphical analysis is as follows:

- For the three average powers $P$, with the increasing laser marking speed, a decrease in the roughness can be observed for $\tau = 100$ ns and $\tau = 200$ ns.
- For the pulse duration of $\tau = 100$ ns: At the power $P$ of 4.72 W with a change in speed from 10 mm/s to 80 mm/s, the roughness changes from 5.5 μm to 0.38 μm. At the power of 6.61 W with a change in speed from 10 mm/s to 80 mm/s, the roughness $R_a$ changes from 5.99 μm to 0.38 μm. At the power of 9.31 W with a change in speed $v$ from 10 mm/s to 80 mm/s, the roughness $R_a$ changes from 7.65 μm to 0.38 μm.
- For the pulse duration of $\tau = 200$ ns: At the power $P$ of 10 W with a change in speed $v$ from 10 mm/s to 80 mm/s, the roughness $R_a$ decreases from 6.33 μm to 4.87 μm. At the power $P$ of 14 W with a change in speed $v$ from 10 mm/s to 80 mm/s, the roughness $R_a$ increases from 9.5 μm to 6.33 μm. At the power $P$ of 19.4 W with a change in speed $v$ from 10 mm/s to 50 mm/s, the roughness $R_a$ increases from 14.5 μm to 10.2 μm.
- When compared with the results of $\tau = 100$ ns pulse duration, it was noticed that the roughness $R_a$ obtained at 200 ns is more stable, rather than the downfall of the achieved graph at 100 ns. The reason is the larger pulse energy at a $\tau = 200$ ns pulse duration than that at $\tau = 100$ ns. At high speeds $v$ of 70–80 mm/s, the roughness $R_a$ of the marked surface approaches that of the background for $\tau = 100$ ns.

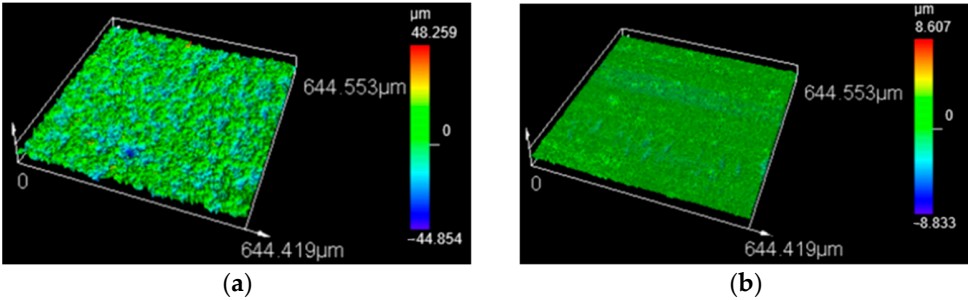

|  |  |
|:-:|:-:|
| (**a**) | (**b**) |

**Figure 5.** Images of the comparison of the highest and lowest roughness $R_a$ of copper sample at $\tau = 100$ ns. (**a**) $P = 9.3$ W at $v = 10$ mm/s, and (**b**) $P = 9.3$ W at $v = 80$ mm/s, respectively.

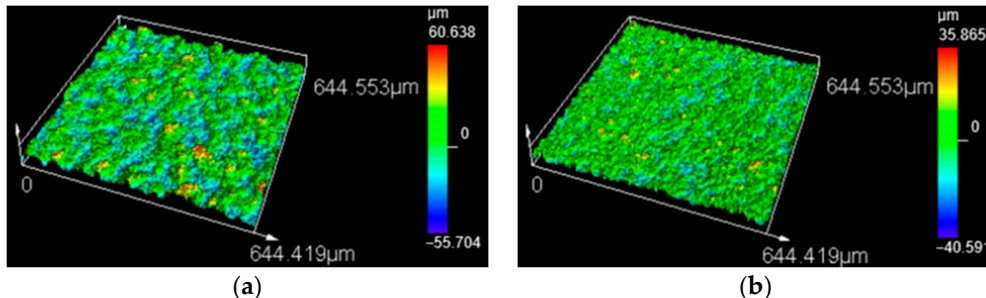

**Figure 6.** Images of the comparison of the highest and lowest roughness of copper sample at $\tau$ = 200 ns. (**a**) $P$ = 19.4 W at $v$ = 10 mm/s, and (**b**) $P$ = 19.4 W at $v$ = 80 mm/s, respectively.

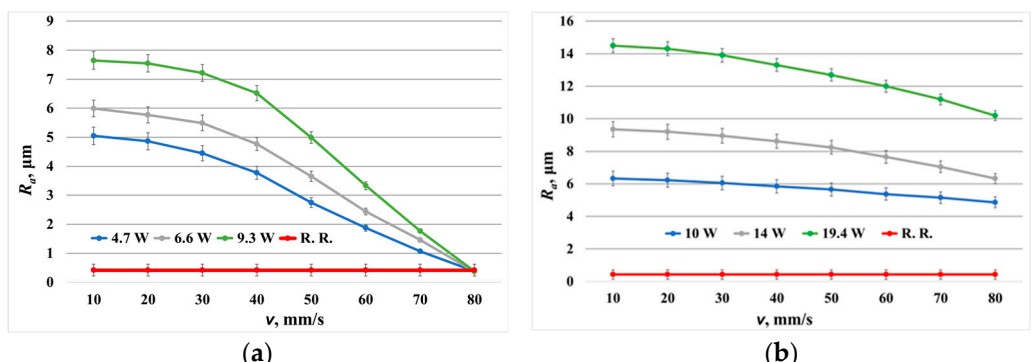

**Figure 7.** Graphical representation of the average roughness measurements of the copper sample, marked with a fiber laser for (**a**) $\tau$ = 100 ns; (**b**) $\tau$ = 200 ns.

*3.2. Determining the Dependence of Microhardness on Speed—HV = HV(v)—For a Fiber Laser for Two Pulse Durations*

The graphical representations in Figure 8 depict the obtained analysis values of the microhardness of Cu after laser processing for a pulse duration of $\tau$ = 100 ns (Figure 8a) and $\tau$ = 200 ns (Figure 8b). The graphical analysis is as follows:

- For the three powers, with the increasing laser marking speed, a decrease in the microhardness can be observed for $\tau$ = 100 ns and $\tau$ = 200 ns.
- For the pulse duration of 100 ns: At the power $P$ = 4.72 W with a change in speed $v$ from 10 mm/s to 80 mm/s, the microhardness $HV$ changes from 179 to 75. At the power $P$ = 6.61 W with a change in speed $v$ from 10 mm/s to 80 mm/s, the microhardness $HV$ changes from 199 to 80. At the power $P$ = 9.31 W with a change in speed $v$ from 10 mm/s to 80 mm/s, the microhardness $HV$ changes from 235 to 84.
- For the pulse duration of 200 ns: At the power $P$ = 10 W with a change in speed $v$ from 10 mm/s to 80 mm/s, the microhardness changes HV from 123 to 82. At the power $P$ of 14 W with a change in speed $v$ from 10 mm/s to 80 mm/s, the microhardness HV changes from 160 to 100. At the power $P$ = 19.4 W with a change in speed $v$ from 10 mm/s to 80 mm/s, the microhardness $HV$ changes from 207 to 123.
- When compared with the results of $\tau$ = 100 ns pulse duration, it was noticed that the enhanced microhardness obtained at $\tau$ = 200 ns is more stable than that at $\tau$ = 100 ns.

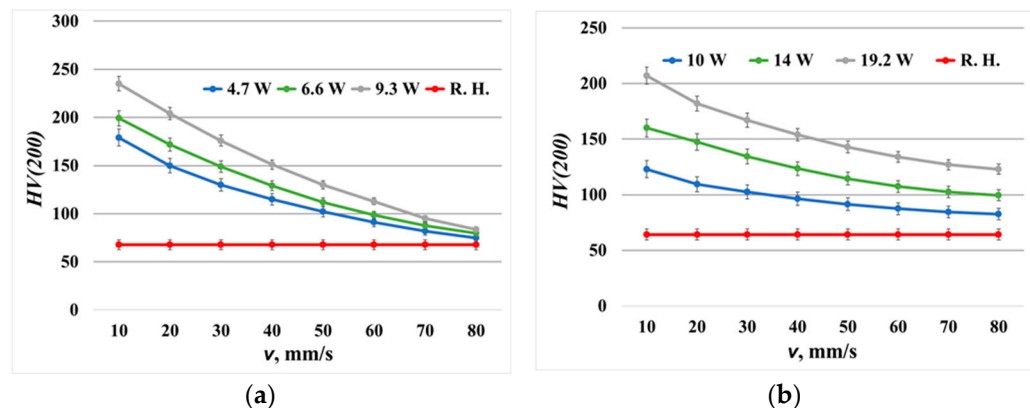

(a)            (b)

**Figure 8.** Graphical representation of the microhardness measurements of the copper plate, marked with a fiber laser for (**a**) $\tau$ = 100 ns; (**b**) $\tau$ = 200 ns.

*3.3. Determining the Dependence of Roughness on Raster Step—$R_a = R_a(\Delta x)$—For a Fiber Laser and a CuBr Laser*

During the experiments, the raster step $\Delta x$ had values of 3 µm, 5 µm, 7 µm, 10 µm, 12 µm, 15 µm, 17 µm and 20 µm for the fiber laser. For the copper bromide laser, the raster step $\Delta x$ values were 3 µm, 7 µm, 12 µm, 17 µm, 22 µm and 27 µm.

The following parameters were kept constant: for the fiber laser—frequency v = 20 kHz, pulse duration 100 ns and power $P$ = 9.3 W; for the CuBr laser—frequency v = 20 kHz, pulse duration $\tau$ = 30 ns and power $P$ = 5.5 W. A comparison of the marked surfaces (highest and lowest roughness) can be observed in Figures 9 and 10. The graphical images in Figure 11 depict the obtained analytical values of the average roughness of the Cu after the laser treatment for the fiber laser (Figure 11a) and the CuBr laser (Figure 11b). The graphical analysis is as follows:

- For the three speeds $v$, with the increasing raster step $\Delta x$, a decrease in the roughness $R_a$ can be observed for the two lasers.
- For the fiber laser: At the speed $v$ of 15 mm/s with a change in raster step $\Delta x$ from 3 µm to 20 µm, the roughness $R_a$ decreases from 30.5 µm to 12.5 µm. At the speed $v$ of 45 mm/s with a change in raster step $\Delta x$ from 3 µm to 20 µm, the roughness $R_a$ decreases from 24.3 µm to 10.3 µm. At the speed $v$ of 90 mm/s with a change in raster step $\Delta x$ from 3 µm to 20 µm, the roughness $R_a$ decreases from 15.6 µm to 7.3 µm.
- For the CuBr laser: At the speed $v$ of 30 mm/s with a change in raster step $\Delta x$ from 3 µm to 27 µm, the roughness $R_a$ changes from 6.4 µm to 2.1 µm. At the speed $v$ of 45 mm/s with a change in raster step $\Delta x$ from 3 µm to 27 µm, the roughness $R_a$ decreases from 5.4 µm to 1.6 µm. At the speed $v$ of 90 mm/s with a change in raster step $\Delta x$ from 3 µm to 27 µm, the roughness $R_a$ decreases from 3.1 µm to 1.0 µm.

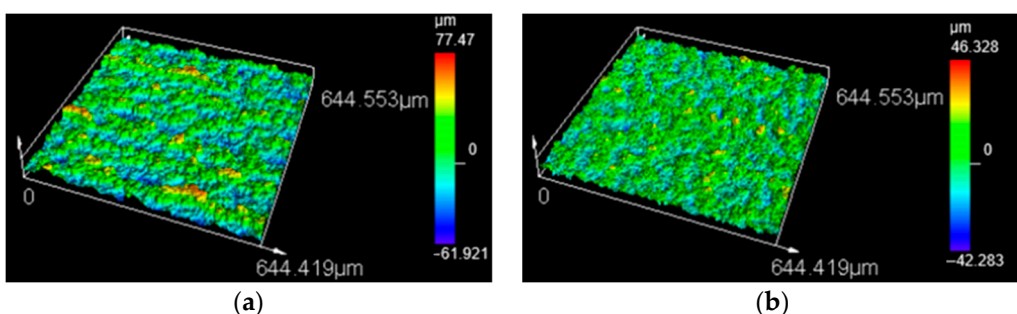

(a)            (b)

**Figure 9.** Images of the comparison of the highest and lowest roughness $R_a$ of the fiber-marked sample at $\tau$ = 100 ns, (**a**) $v$ = 90 mm/s, $\Delta x$ = 3 µm, and (**b**) $v$ = 90 mm/s, $\Delta x$ = 20 µm, respectively.

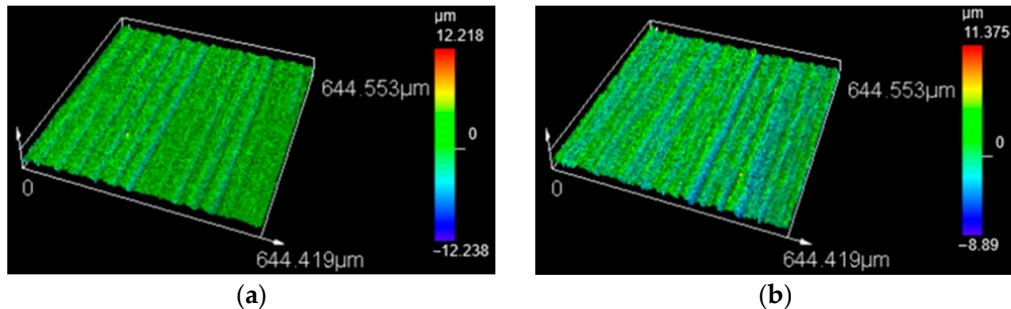

**Figure 10.** Images of the comparison of the highest and lowest roughness $R_a$ of the CuBr marked sample at $\tau$ = 30 ns, (**a**) $v$ = 90 mm/s, $\Delta x$ = 3 μm, and (**b**) $v$ = 90 mm/s, $\Delta x$ = 27 μm, respectively.

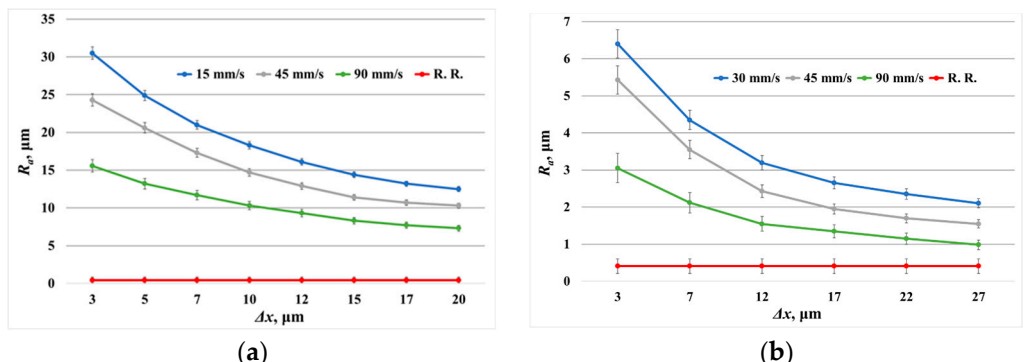

**Figure 11.** Graphical representation of the roughness $R_a$ measurements of the marked sample with (**a**) fiber laser, (**b**) CuBr laser.

*3.4. Determining the Dependence of Microhardness on Raster Step—HV = HV($\Delta x$)—For a Fiber Laser and a Copper Bromide Laser*

The graphical images in Figure 12 depict the obtained analytical values of the microhardness of the Cu after the laser treatment for the fiber laser (Figure 12a) and the copper bromide laser (Figure 12b). The graphical analysis is as follows:

- For the three speeds $v$, with the increasing raster step $\Delta x$, a decrease in the microhardness can be observed for the two lasers.
    - For the fiber laser: At the speed $v$ of 15 mm/s with a change in raster step $\Delta x$ from 3 μm to 20 μm, the microhardness $HV$ decreases from 126 to 95. At the speed $v$ of 45 mm/s with a change in raster step $\Delta x$ from 3 μm to 20 μm, the microhardness HV changes from 143 to 137. At the speed $v$ of 90 mm/s with a change in raster step $\Delta x$ from 3 μm to 20 μm, the microhardness $HV$ increases from 157 to 293.
    - For the CuBr laser: At the speed $v$ of 30 mm/s with a change in raster step $\Delta x$ from 3 μm to 27 μm, the microhardness $HV$ decreases from 320 to 165. At the speed $v$ of 45 mm/s with a change in raster step $\Delta x$ from 3 μm to 27 μm, the microhardness $HV$ changes from 273 to 130. At the speed $v$ of 90 mm/s with a change in raster step $\Delta x$ from 3 μm to 20 μm, the microhardness $HV$ increases from 216 to 112.
- Microhardness when marking with a CuBr laser is about 20% greater than when using a fiber laser. The reason is that the pulse power $P_p$ for a copper bromide laser is 9.17 kW, while for a fiber laser it is only $P_p$ = 4.65 kW.
- For both lasers, microhardness $HV$ gradually decreases with increasing speed $v$ and raster step $\Delta x$.

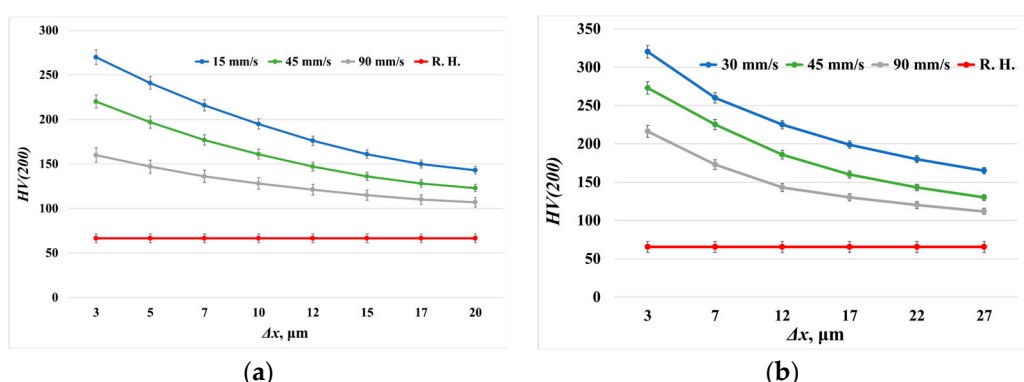

**Figure 12.** Graphical representation of the microhardness *HV* measurements of the marked sample with (**a**) fiber laser, (**b**) CuBr laser.

## 4. Conclusions

The results show that the surface roughness $R_a$ and microhardness *HV* for laser marking are dependent on laser parameters, like power *P*, pulse duration $\tau$, scanning speed *v* and raster step $\Delta x$. It was noticed that with the increase in the power *P* and pulse duration $\tau$, the roughness $R_a$ and microhardness *HV* increased too, whereas, with the increase in the scanning speed *v* and raster step $\Delta x$, a decrease in the values of roughness $R_a$ and microhardness *HV* were obtained.

Also, results were obtained for the impact of basic technological parameters, such as scanning speed *v* and raster step $\Delta x$, to optimize the process of laser marking of copper specimens with various lasers—a fiber laser and a CuBr laser. Specific values for the change in roughness and microhardness in the studied ranges of variation in the technological parameters were also obtained from the dependency graphs and analyses:

- The surface roughness $R_a$ and microhardness *HV* from speed *v* for three powers of laser radiation at pulse duration $\tau$ = 100 ns for the fiber laser. For the studied speed interval from 10 mm/s to 80 mm/s, it was found that with increasing speed, the roughness sharply decreases for all three powers, as the roughness changes from 5.50–7.65 μm to 0.38 μm, i.e., decreases 15–20 times. The microhardness decreases about 2.5 times in the studied speed interval.
- The surface roughness $R_a$ and microhardness *HV* from speed *v* for three powers of laser radiation at pulse duration $\tau$ = 200 ns for the fiber laser. Again, as the speed increases, the roughness decreases, but the reduction is about 30% for the whole range of speeds, i.e., significantly slower compared to the pulse duration $\tau$ = 100 ns. It is found that the microhardness at pulse duration $\tau$ = 100 ns is about two times greater than that at pulse duration $\tau$ = 200 ns.
- The surface roughness $R_a$ and microhardness *HV* from the raster step $\Delta x$ for three speeds *v* for the fiber laser. When changing the raster step from 3 μm to 20 μm, the roughness decreases by 2.5 times, and the microhardness decreases by nearly two times.
- The surface roughness $R_a$ and microhardness *HV* from the raster step for three speeds *v* for the CuBr laser at $\tau$ = 30 ns. When changing the raster step from 3 μm to 27 μm, the roughness decreases by three times, and the microhardness decreases by about 1.9 W times.

The results are suitable for obtaining a complete picture of the laser marking process and can be applied to the analysis of other technological processes related to laser surface treatment. The results are suitable for the field of mechanical engineering and the military industry, where surface treatments and tribological characteristics are of utmost importance.

**Author Contributions:** Conceptualization—L.L. and E.Y.; methodology—E.Y. and R.S.G.; validation—L.L., E.Y. and R.S.G.; formal analysis—L.L., N.A., E.Y. and R.S.G.; investigation—R.S.G. and E.Y.;

data curation—N.A., R.S.G., E.Y. and L.L.; writing—original draft preparation—R.S.G., E.Y., N.A. and L.L.; writing—review and editing—L.L., E.Y. and N.A.; visu-alization—E.Y., N.A. and R.S.G.; supervision—L.L.; project administration—E.Y. and L.L.; funding acquisition—E.Y. All authors have read and agreed to the published version of the manuscript.

**Funding:** The authors gratefully acknowledge the financial support of the European Regional Development Fund, Postdoctoral research aid Nr. 1.1.1.2/16/I/001 research application "Analysis of the parameters of the process of laser marking of new industrial materials for high-tech applications, Nr. 1.1.1.2/VIAA/3/19/474".

**Institutional Review Board Statement:** Not applicable.

**Informed Consent Statement:** Not applicable.

**Data Availability Statement:** Additional data from our research can be found in "Proceedings of the International Scientific and Practical Conference", http://journals.rta.lv/index.php/index.

**Conflicts of Interest:** The authors declare no conflict of interest.

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
