# Peer review of "Investigation of the Change in Roughness and Microhardness during Laser Surface Texturing of Copper Samples by Changing the Process Parameters"

_coatings, doi:10.3390/coatings13111970_

Round 1

Reviewer 1 Report

Comments and Suggestions for Authors

The manuscript is well written and can be accepted in its current form

Author Response

Thanks for the suggestion. We reviewed the entire manuscript and added additions to the reviewers' requests related to the analyzes of results and conclusions.

Reviewer 2 Report

Comments and Suggestions for Authors

1. What is the objective of this study.

2. On what basis the Texturing is selcted.

3. The authors  have not discussed pattern , geometry and size  of the pattern used .

4. The surface  morphological studies of the samples such as FESEm and EDS are not given after and before texturing.

5. What is the depth of the dimples used ..

6. Authors may carry some further tests of friction and wear to make this paper more suitable.

7. How is this texturing related to coating.

8. Figures 9-10 are not legible and need sto be repalce .

9 Photographs of machines to be removed.

10 Researchers  need to discuss the model of surface roughness 

11. More experimental studies need to be incorporated 

Comments on the Quality of English Language

NOt relevant

Author Response

-  What is the purpose of the present study?

We've adjusted the purpose of the present study based on your feedback. We believe that this variant is more clear and specific and corresponds to the research we have done.

The article aims to investigate the laser marking process of copper samples and analyze the influence of some basic parameters such as speed, raster step, laser radiation power and the influence of wavelengths for two types of lasers (fiber laser and copper bromide laser) on roughness and microhardness.

-  On what basis is the Texturing selected?

Based on experience gained over several years and our studies on laser marking of copper with both types of lasers have not been conducted. The marking of copper materials is accompanied by a change in surface roughness and hardness according to the laser marking mode.

-  The authors did not discuss the model, geometry and size of the model used.

The matrix version of the experiment makes it possible to determine the change of roughness and hardness depending on the variation of the laser marking modes, the raster step, speed, pulse duration and average power. The dimensions of the matrices were selected to provide sufficient surface area for roughness and hardness analysis with the selected experimental apparatus. The selected experimental matrix is described in detail in the new edited version of the report.

-  Surface morphological studies of the samples such as FESEm and EDS are not given after and before texturing.

The tests carried out on the samples before processing are added at the beginning of the results point.

-  What is the depth of the dimples used?

The marking depth is not investigated. The change in roughness Ra (average roughness - obtained from troughs and ridges for a length of 644 µm) was investigated.

-  Authors can carry some additional friction and wear tests to make this paper more suitable.

Thanks for the proposed tribological studies. At this stage, they have not been conducted, but the team of authors has already discussed conducting them in order to establish their behavior according to the laser marking regime.

-  How is this texturing related to coverage?

The title, abstract and introduction do not refer to coating, but to surface texturing, surface modification by laser processing.

-  Figures 9-10 are not readable and should be repalced .

Figures have been corrected in larger font for easy reading.

-  Pictures of machines needs to be removed.

The motion was accepted and they were removed..

-  Researchers should discuss the pattern of surface roughness

We have added a description in the experiment methodology for the roughness model. It is justified why the roughness Ra was chosen as an indicator for the investigated laser marking regimes.

  • More experimental studies should be included

The proposed studies are too many because of the number of markings with different laser marking regimes. The speed, raster step and pulse duration have been changed. Studies were conducted with two types of lasers with different wavelengths. The results are presented for comparison between them. As a result, roughness and hardness were measured for each marking square. Graphical dependencies have been built showing in which direction they change depending on the laser marking regime. Of course, more experimental studies can be done, but the limit of pages per publication is limited.

Reviewer 3 Report

Comments and Suggestions for Authors

In this manuscript the authors report on the change of roughness and microhardness during laser surface texturing of copper samples by changing the process parameters. Although the authors conducted sufficient survey in the introduction, it was only a compilation of literature and did not summarize information. The manuscript lacks analysis of the results and is more like an experimental report. Therefore, I recommend the rejection in the present form. 

Comments on the Quality of English Language

The quality of English Language is needed to be improved.

Author Response

In response to your review, we may note the following:

  • A clear and specific goal of the study is set in the manuscript;
  • There is a developed methodology for designing and performing the experiments;
  • The influence of speed, raster step and power on the roughness and microhardness of the resulting textured surfaces was investigated;
  • The results for pulse durations of 100 ns and 200 ns were compared and analyzed;
  • The results for two wavelengths (fiber laser and CuBr laser) were compared.

Considering that some results could be interpreted in more detail, we performed an additional analysis.

Reviewer 4 Report

Comments and Suggestions for Authors

Dear Editor,

Dear Authors

of the article entitledInvestigation of the change of roughness and microhardness 2 during laser surface texturing of copper samples by changing 3 the process parameters”.

The article I received for review concerns issues that in recent years have strongly exploited by many scientists from many fields. No wonder, because both the technology of surface texturing itself, as well as the use of lasers for this purpose, are an important element of modern materials science. However, the research itself is very basic, and I would rather expect a much richer work by publishing in such a journal. I wonder if the section and theme of the special issue really correspond to the content of the article, but here the decision is of course up to the editor.

In the introduction, the authors extensively discuss theoretical issues, recalling research, methodology, conclusions and observations of other scientists dealing with this topic. Pointing out the similarities and differences between them. Importantly, most of the cited articles were published in the last two or three years, which indicates the novelty of the research.

Methodology - theoretically everything is here, but described in a shallow, not very scientific way.

Results - the authors described the obtained results correctly, but they are difficult to read, because you have to go back every now and then in order to compare the results. I would suggest that, in addition to the description, the authors add a table with the results taking into account all the variables - then it will be much easier for the reader to analyze the research results.

Conclusions – the most important observations are contained therein. I have the impression that the article is written a bit sloppy, inattentively - which is puzzling, especially when I analyzed the latest publications of Lyubomir Lazov and Emil Yankov. Gentlemen, this year they wrote a lot of really good articles, so it may be worth using a proven scheme and also correcting the shortcomings, which are quite a lot in this work

Mistakes:

A colon instead of a period at the end of a sentence - line 33

Semicolon instead of comma - line 71

Units - sometimes authors use a space between value and unit, sometimes not - example line 149/156; 133-156

Bullets after colons made of lowercase letters, separated by commas - line 176/185, same for 238, 272, 302, etc.

Caption Fig 3./Fig 6.

Not separated from the rest of the description I consider Fig. 4/5/6 redundant

Manufacturers, place of production of the measuring equipment used for testing ??

Empty bullet - line 256, 315

The graphs should be larger, they are illegible at this scale

[22] [23] – redundant

Author Response

Thanks for the review. Corrections have been made regarding the English language.

  • Methodology - theoretically everything is here, but described in a shallow, not very scientific way.

The methodology has been modified and described in more detail.

  • Conclusions

The conclusions have been revised. They show the achieved objectives and experimental results of the report.

Errors:

  • Colon instead of period at the end of the sentence - line 33 – corrected remark;
  • Semicolon instead of comma - row 71 – corrected remark;
  • Units - sometimes authors use a space between value and unit, sometimes not - example line 149/156; 133-156 – identified inaccuracies have been corrected;
  • Bullets after colons made of lowercase letters, separated by commas - line 176/185, same for 238, 272, 302, etc. – the detected errors have been fixed.
  • Caption Fig 3./Fig 6. - the figure text has been added
  • Not separated from the rest of the description I consider Fig. 4/5/6 redundant – figures 4/5/6 are removed. The methodology of the experiment is described in detail.
  • Manufacturers, place of production of the measuring equipment used for testing ?? – The scientific studies describe the name and model of the equipment. I have not come across a description of the manufacturer, place of manufacture, year of manufacture. The articles are not intended to provide such information as to advertise the manufacturer.
  • Empty bullet - line 256, 315 – There is always one line indent between the main text and a figure.
  • The graphs should be larger, they are illegible at this scale – fixed the graphics with a larger and readable size of the scales and the legend.
  • [[22] [23] redundant – references [22] and [23] have been removed after the removal of figures 4/5/6.

Round 2

Reviewer 2 Report

Comments and Suggestions for Authors

Authors have not included the Hv images . Pl include HV image sin this paper 

and details of HV tests.

Hv vs indentation time etc

Author Response

Added additional comments and figures for measured hardness. Figures are shown of the measured hardness of the material as supplied, the machined surface with overlap, and the measured hardness without overlap. The figures provided show the effect of marking speed and raster pitch on roughness and hardness.

Reviewer 3 Report

Comments and Suggestions for Authors

Compared to the original manuscript, the revised manuscript has hardly made significant modifications. The Results section is still a list of experimental data without discussion. Different experimental parameters inevitably lead to different experimental results, but what are the mechanisms under these differences results? The Conclusion section also lacks valuable items.

Author Response

Thank you for your recommendations, they are valuable to us. Changes have been made to all points marked in yellow since the first revision of the publication. We reviewed the entire manuscript and added additions to the reviewers' requests related to the analyzes of results and conclusions.

Regarding point results, comparisons and analyzes are presented in an algorithm following the form of the experiment. The results of the marking with the two lasers are compared for the variation of roughness and hardness depending on the speed and raster step. They have an impact on the overlap coefficient, which also leads to changes in roughness and hardness.

According to the research results, it can be concluded that the changing parameters of laser processing can be controlled with these characteristics of the processed surface, such as roughness and microhardness.

Reviewer 4 Report

Comments and Suggestions for Authors

Accept in present form

Author Response

(The authors gave the same response as above.)
